# Inhibition of SGLT2 Preserves Function and Promotes Proliferation of Human Islets Cells In Vivo in Diabetic Mice

**DOI:** 10.3390/biomedicines10020203

**Published:** 2022-01-18

**Authors:** Daniel Karlsson, Andrea Ahnmark, Alan Sabirsh, Anne-Christine Andréasson, Peter Gennemark, Ann-Sofie Sandinge, Lihua Chen, Björn Tyrberg, Daniel Lindén, Maria Sörhede Winzell

**Affiliations:** 1Bioscience Metabolism, Research and Early Development, Cardiovascular, Renal and Metabolism (CVRM), BioPharmaceuticals R&D, AstraZeneca, Mölndal, 43150 Gothenburg, Sweden; Daniel.Karlsson@astrazeneca.com (D.K.); andrea.ahnmark@astrazeneca.com (A.A.); anne-christine.andreasson@astrazeneca.com (A.-C.A.); chen_lihua@grmh-gdl.cn (L.C.); daniel.linden@astrazeneca.com (D.L.); 2Advanced Drug Delivery, Pharmaceutical Sciences R&D, AstraZeneca, Mölndal, 43150 Gothenburg, Sweden; alan.sabirsh@astrazeneca.com; 3Drug Metabolism and Pharmacokinetics, Research and Early Development, Cardiovascular, Renal and Metabolism (CVRM), Biopharmaceuticals R&D, AstraZeneca, Mölndal, 43150 Gothenburg, Sweden; peter.gennemark@astrazeneca.com (P.G.); ann-sofie.sandinge@astrazeneca.com (A.-S.S.); 4Department of Biomedical Engineering, Linköping University, 58183 Linköping, Sweden; 5Translational Science and Experimental Medicine, Research and Early Development, Cardiovascular, Renal and Metabolism (CVRM), Biopharmaceuticals R&D, AstraZeneca, Mölndal, 43150 Gothenburg, Sweden; bjorn.kiros@outlook.com; 6Department of Physiology, Institute of Neuroscience and Physiology, Sahlgrenska Academy, University of Gothenburg, 41390 Gothenburg, Sweden

**Keywords:** human islets, transplantation, insulin, glucagon, dapagliflozin, diabetes, transdifferentiation, alpha cells, beta cells

## Abstract

Dapagliflozin is a sodium-glucose co-transporter 2 (SGLT2) inhibitor used for the treatment of diabetes. This study examines the effects of dapagliflozin on human islets, focusing on alpha and beta cell composition in relation to function in vivo, following treatment of xeno-transplanted diabetic mice. Mouse beta cells were ablated by alloxan, and dapagliflozin was provided in the drinking water while controls received tap water. Body weight, food and water intake, plasma glucose, and human C-peptide levels were monitored, and intravenous arginine/glucose tolerance tests (IVarg GTT) were performed to evaluate islet function. The grafted human islets were isolated at termination and stained for insulin, glucagon, Ki67, caspase 3, and PDX-1 immunoreactivity in dual and triple combinations. In addition, human islets were treated in vitro with dapagliflozin at different glucose concentrations, followed by insulin and glucagon secretion measurements. SGLT2 inhibition increased the animal survival rate and reduced plasma glucose, accompanied by sustained human C-peptide levels and improved islet response to glucose/arginine. SGLT2 inhibition increased both alpha and beta cell proliferation (Ki67^+^glucagon^+^ and Ki67^+^insulin^+^) while apoptosis was reduced (caspase3^+^glucagon^+^ and caspase3^+^insulin^+^). Alpha cells were fewer following inhibition of SGLT2 with increased glucagon/PDX-1 double-positive cells, a marker of alpha to beta cell transdifferentiation. In vitro treatment of human islets with dapagliflozin had no apparent impact on islet function. In summary, SGLT2 inhibition supported human islet function in vivo in the hyperglycemic milieu and potentially promoted alpha to beta cell transdifferentiation, most likely through an indirect mechanism.

## 1. Introduction

Chronic hyperglycemia and impaired beta cell function are hallmarks of diabetes. Reducing the blood glucose increases the functional beta cell mass in diabetic patients and in rodent models of type 2 diabetes [1,2,3,4]. In type 1 diabetes, the continuous loss of beta cells results in severe hyperglycemia at diagnosis. However, after normalization of glucose with insulin treatment, many patients experience a remission phase whereby blood glucose is easily controlled with low doses of injected insulin [5]. Thus, relieving the islets of the high glucose burden has been shown to preserve functional beta cells [1,2,3,4]. Restoration of functional beta cell mass is an attractive concept for the treatment of (and ideally curing) diabetes mellitus. Several potential mechanisms, including beta cell proliferation, reduced apoptosis, neogenesis, re-differentiation, and transdifferentiation of other pancreatic cell types into beta cells and cell therapy, have been suggested as targets for drug development [6,7,8,9,10,11,12]. Mechanisms of beta cell regeneration have been studied in preclinical experimental models of severe beta cell loss induced by pharmacological destruction or diphtheria toxin-induced beta cell ablation [8,13,14,15]. Following near-total ablation of beta cells in mice, it was demonstrated that new beta cells could arise from existing alpha cells [8,13,14,15]. Interestingly, alpha cell ablation gives a weak phenotype with normoglycemia and maintains beta cell function and near-normal glucagon levels [16]. Alpha and beta cells are closely related cell types, and by modifying the expression of a few transcription factors such as PDX-1, ARX, and PAX4, the cells can transform from one to the other [7,17,18]. In patients with diabetes mellitus of both type 1 and type 2, alpha cell transdifferentiation into beta cells could represent a potential mechanism for regeneration of the functional beta cell mass. However, the mechanisms that trigger alpha to beta cell transdifferentiation in human islets are much less studied. Recently, new evidence for human islet cell plasticity was presented, where it was demonstrated that human alpha cells can turn into glucose-dependent insulin-secreting cells [19].

Sodium-glucose co-transporter 2 (SGLT2) inhibitors represent a relatively novel glucose-lowering drug class [20], which has been shown to be safe and efficacious also in type 1 diabetic patients [21,22]. SGLT2 inhibition blocks glucose resorption in the kidney proximal tubule and promotes glucosuria, thereby decreasing blood glucose levels. There are conflicting data as to whether SGLT2 is expressed in alpha cells, and recent studies have proposed that the effect observed on glucagon secretion is more likely a result of indirect paracrine signaling between the different islet cells [23,24]. However, recent clinical studies in type 2 diabetic patients revealed protective effects on beta cell function in addition to improved insulin sensitivity following treatment with SGLT2 inhibitors [3,25]. Although several studies in rodents have demonstrated that SGLT2 deletion and pharmacological inhibition of SGLT2 preserve beta cell function [26,27,28,29,30,31], little is known about the mechanisms, particularly in human islets. There are distinct structural and functional differences between rodent and human islets, highlighting the importance of conducting pharmacological studies with primary human islets to better predict the clinical outcome [32,33,34]. We used an alloxan-induced type 1-like diabetic mouse model transplanted with human islets to explore the effects of the selective SGLT2 inhibitor dapagliflozin on the function of grafted human islets in vivo. In addition, we measured the alpha and beta cell composition and the number of proliferating cells using triple-staining immunohistochemistry methods. In vitro experiments with human islets do not support a direct effect of dapagliflozin for the observed in vivo effects. This study shows that dapagliflozin protects human islet function in vivo in the diabetic milieu and provides initial support for alpha to beta cell transdifferentiation as a potential way of increasing functional beta cell mass in diabetic human islets.

## 2. Materials and Methods

### 2.1. Animals

Immune-compromised male NUDE (Crl:NMRI-Foxn1nu) mice (Charles River, Freiburg, Germany), 8 weeks of age, were singly housed in autoclaved, individually ventilated cages under controlled temperature (21 ± 1 °C) and humidity (50% ± 10%) with a 12 h light/dark cycle (06:00–18:00 light). The mice had free access to water and food (R70; Lactamin, Kimstad, Sweden). Body weight was monitored three times per week or every day if the body weight was decreasing. Individuals that showed signs of affected health or lost more than 10% of their initial body weight two consecutive days were excluded from the study in accordance with the ethical experimental protocol. Only those mice that completed the 8-week study were included in the final analysis.

### 2.2. Human Islets, Islet Transplantation, and Diabetes Induction

Human pancreatic islets were obtained from Prodo Laboratories (Aliso Viejo, CA, USA). Before transplantation, the islets were incubated for 24–48 h in complete Prodo Islet Media Standard (PIMS). Islets from three normoglycemic donors (Appendix A) were used, and 1500 islet equivalents (IEQs) were transplanted into the left kidney sub-capsular space of the mice representing a marginal islet mass as previously described [35]. A total of 19–20 animals were transplanted per islet donor in three separate experiments performed in one-week intervals. Islet grafts were allowed to integrate into the host tissue for 2 weeks before diabetes was induced by ablating the mouse beta cells using a single intravenous injection of alloxan monohydrate (72.5 mg/kg, Sigma-Aldrich, Dorset, UK). Normoglycemic sham recipients went through the full surgical procedure and received an equivalent volume of saline under the kidney capsule.

### 2.3. Dapagliflozin Treatment

Five days post diabetes induction, the mice were randomized into two groups per islet donor based on non-fasting glucose levels. Mice were given drinking water with or without dapagliflozin (~3 mg/kg/day) for 53 days. Plasma exposure to the drug was determined regularly. Sham recipients served as non-diabetic controls and were not treated with dapagliflozin. Dapagliflozin is the active component of a registered drug, and it was synthesized by AstraZeneca and released for use after completion of standard release testing, assuring the right structure and activity of the compound.

### 2.4. Pharmacokinetic Analysis of Dapagliflozin Administered in the Drinking Water

The prescribed daily dose of dapagliflozin for humans is 5 or 10 mg. A pooled analysis across 20 studies in healthy subjects and in those with T2DM reports the corresponding area under the dapagliflozin plasma concentration-time curves (AUCs; 0–24 h) at a steady state as 290 ng × h/mL (or 0.71 µM × h; molecular weight = 409 g/mol) and 650 ng × h/mL (1.6 µM × h), respectively [36]. The 5 and 10 mg doses result in ~55% and ~70% of maximum change from baseline in 24 h urinary glucose [36]. The plasma protein binding of dapagliflozin is 93% and 91% in mice and humans, respectively ([37] and internal data). We designed the mouse study to have a therapeutically relevant unbound dapagliflozin plasma exposure. Hence, we aimed for a total AUC in the interval 0.91–2.1 µM × h (calculated as AUC_human_ × fu_human_/fu_mouse_, where fu = fraction unbound).

Prior to the study, we investigated the pharmacokinetics in immune-compromised male NUDE mice by dosing 3 mg/kg/day dapagliflozin in drinking water to 6 animals. The AUC was estimated to 1.3 µM × h (CV = 15%; non-compartmental analysis). Hence, the AUC of the 3 mg/kg/day dose fell in the desired interval of 0.91–2.1 µM × h, and the dose was selected for the study. Due to the dependency of water intake on the diabetic state of the mice, the average actual dose was 3.7 ± 0.2 mg/kg/day during the study period. Assuming linear pharmacokinetics, the predicted AUC in the study was 1.6 µM × h, hence in the upper end of the desired interval.

### 2.5. Bioanalysis of Dapagliflozin

All blood samples were quantified for dapagliflozin content using an Acquity UPLC^®^ I-class system and a Xevo^TM^ TQ-S triple quadrupole mass spectrometer (Waters Corporation, Milford, MA, USA). As analytical column, an ACQUITY UPLC^®^ BEH C18, 2.1 × 50 mm column was used with a 1.7 µm particle size maintained at 40 °C. Mobile phase A consisted of Milli-Q water with 2% acetonitrile and 10 mmol/L ammonium acetate, pH 5, and mobile phase B consisted of acetonitrile with 10 mmol/L ammonium acetate, pH 5. The instrument was operating in negative electrospray ionization mode. The lower limit of quantification was 0.0104 µmol/L.

### 2.6. Intravenous Arginine/Glucose Tolerance Test (IVArgGTT)

To determine the maximal insulin secretory capacity of the beta cells, an IVArgGTT was performed 10–16 days before termination. The animals were fasted for 2 h prior to an intravenous injection of glucose and arginine (350 mg/kg glucose and 250 mg/kg arginine). Blood samples were taken from the tail vein at 0, 1, and 10 min after the injection. Plasma glucose was determined using a glucometer (AccuCheck mobile^®^, Roche Diagnostics, Solna, Sweden) and insulin using the Ultra-Sensitive Mouse Insulin ELISA kit (Chrystal Chem Inc., Elk Grove Village, IL, USA).

### 2.7. Tissue and Blood Sampling and Biochemical Analysis

Non-fasting plasma glucose levels were measured in tail vein blood (Accu-Chek mobile^®^) once weekly. At days 0, 28, and 53, human C-peptide was analyzed (hC-peptide ELISA, Mercodia, Uppsala, Sweden). After 53 days of treatment, a final blood sample was collected by cardiac puncture under isoflurane (Forene^®^, Abbot Scandinavia, Solna, Sweden) anesthesia, and the mice were euthanized by heart dissection. Blood samples, collected in EDTA tubes, were centrifuged at 4000× *g* for 10 min, and plasma was stored at −20 °C until analysis. Metabolic markers were measured in the plasma samples: colorimetric kit methods were used to analyze alanine aminotransferase (ALT CP, Horiba ABX SAS, Montpellier, France), total cholesterol (Cholesterol CP, Horiba ABX SAS), haptoglobin (Phase^TM^ Range Haptoglobin, Colometric Assay, Tridelta Development Limited, Kildare, Ireland), fructosamine (Fructosamine, Horiba ABX SAS), triglycerides (Trig/GB, Roche Diagnostics GmbH, Mannheim, Germany) and 3-D-hydroxybutyrate (RANBUT: D-3-hydroxybutyrate, Randox laboratories limited, Crumlin, United Kingdom). Analyses were performed using an ABX Pentra 400 (HORIBA Medical, Irvine, California, USA). Triglycerides were extracted from liver biopsies and insulin from pancreas biopsies by incubation in acidic ethanol. The human islet grafts and the mouse pancreas were dissected and fixed in 4% formalin.

### 2.8. Histology and Immunohistochemistry

After formalin fixation, dehydration, and paraffin embedding, the islet grafts were sectioned and mounted on three slides to enable multiple staining protocols. Multiplexed automated staining was performed using a Ventana Ultra system (Ventana Medical Systems, Inc. Roche Group, Oro Valley, AZ, USA). Primary antibodies used were specific for Ki67 (Abcam, ab15580), cleaved caspase-3 (Biocare Medical, CP299A), PDX-1 (Abcam, ab47267), insulin (USBio, I7660-24A), and glucagon (Sigma, G2654, monoclonal). Antibody binding was visualized using DAB (3,3-diaminobenzidine tetrahydrochloride), purple and yellow chromogens, respectively, using the biotin-free detection system designed for the Ventana system (OmniMap anti-Rb HRP, Ventana Medical Systems). Hematoxylin was added as a nuclear counter-stain.

Slides from mouse pancreas were embedded according to Paulsen et al. [38], and triple IHC stained using the Intellipath (Biocare Medical, Pacheco, CA, USA) IHC staining system with antibodies against insulin, glucagon, and Ki67 using reagents provided by Biocare.

Three levels (100 µm apart) of each islet graft were analyzed using Visiopharm Integrator System software (version 2017.7, Visiopharm, Hørsholm, Denmark). To analyze the grafted areas, the digital slides were resampled to create lower resolution images, and machine learning, combined with Bayesian classification, was used to identify first the entire tissue followed by regions of interest around each graft. These regions were manually optimized if necessary. Within these regions the slides were again analyzed, using the maximum resolution.

Insulin/glucagon/Ki67 triple-staining images were analyzed, and data are presented as percentage of all insulin and glucagon-positive cells detected in the images as well as the percentage of double-positive Ki67^+^insulin^+^ and Ki67^+^glucagon^+^ cells to determine the percentage of proliferating cells. PDX-1-positive nuclei were identified, and a subpopulation of these nuclei with at least 50% of their perimeter in contact with glucagon-positive areas was also created. Results are presented as the percentage of dual-positive PDX-1^+^/glucagon^+^ cells within the entire population of glucagon^+^ cells.

Apoptotic endocrine cells were detected by staining the human islet grafts for caspase 3 in combination with insulin and glucagon. The area of insulin and glucagon staining was outlined manually and measured using BioPix. The number of double-positive insulin/caspase 3 and glucagon/caspase 3 cells were manually counted at 40× magnification. The analysis was performed twice.

For mouse pancreas tissue sections, insulin and glucagon stained areas and the total pancreas area were measured using BioPix (BioPix, Göteborg, Sweden) image analysis software.

### 2.9. Human Islets Culture and Dapagliflozin Treatment

Human islets were obtained from three non-diabetic donors (Prodo Laboratories, Inc., Aliso Viejo, CA, USA, Appendix A) and maintained in standard Prodo islet media (PIM(S)) containing 5% human AB serum, PIM(G), and 1% penicillin/streptomycin (P/S). One day after arrival, the media was replaced with RPMI1640 containing 5.6 mM glucose, 1 mM glutamine, and 1% P/S.

Human islets were divided into six wells and cultured in RPMI1640 media containing 1, 15, or 30 mM glucose with and without the addition of 100 nM dapagliflozin for 96 h. Media was changed every 24 h. After the 96 h incubation, islets were handpicked into Krebs-Ringer HEPES buffer (KRH; NaCl 130 mmol/L, KH_2_PO_4_ 1.25 mmol/L, MgSO_4_ 1.25 mmol/L, CaCl_2_ 2.68 mmol/L, NaHCO3 5.26 mmol/L, and HEPES 10 mmol/L, pH 7.4) with 0.1% non-fatty BSA for glucagon and insulin secretion assays. Islets were washed twice with 37 °C pre-warmed KRH buffer without glucose and then incubated with KRH buffer without glucose for 60 min. The assay was performed by incubating 4 islets/vial (triplicates per condition) with 200 µL KRH buffer containing 1, 5, 15, or 30 mM glucose for 60 min at 37 °C. After the incubation, 100 µL supernatant was collected, and insulin and glucagon secretion were determined by specific ELISA kits (Mercodia, Uppsala, Sweden).

### 2.10. Statistical Analysis

Area measurements of IHC-stained tissue sections are presented as median with error bars representing the interquartile range as the data were logarithmically distributed. Statistical differences between groups were determined using the Mann–Whitney non-parametric test. The in-life data parameters are presented as the mean and standard error of the mean (SEM) unless otherwise stated. Comparisons between three groups were performed using one-way ANOVA, followed by Turkey’s post-hoc test. Comparisons between two groups were performed using an unpaired, two-tailed Student *t*-test. For time-course data, two-way ANOVA was used, followed by Bonferroni post-hoc test. A *p*-value of *p* ˂ 0.05 was considered statistically significant. Statistical analyses were performed using GraphPad Prism 7.02 software.

## 3. Results

### 3.1. SGLT2 Inhibition Maintained Long-Term Glycemic Control in Diabetic Mice

Alloxan-diabetic mice (*n* = 59) transplanted with human islets were given dapagliflozin (*n* = 25) or tap water (*n* = 34). The average dose was 3.7 ± 0.2 mg/kg/day, and the mice were exposed to dapagliflozin at therapeutically relevant concentrations. Six normoglycemic, sham-operated mice were used to illustrate the non-diabetic phenotype. The body weight curves for the mice completing the study did not differ between the three groups (Figure 1A). Over the 8 weeks study, body weight increased slightly in all three groups with no difference between treatment groups with drug or with the controls that received no drug and no transplant and were non-diabetic. This shows that the mice in all groups that completed the study were in suitable condition, maintaining their body weight. Mice were terminated and removed from the study if the body weight loss was more than 10% for two consecutive days, compared to the starting weight. In the diabetic control group, 17 mice were not able to complete the 8-week study, while only 4 mice on dapagliflozin were terminated before study completion (Figure 1B). Blood glucose in the diabetic mice before starting treatment averaged 15.5 ± 1.5 mM. Human insulin levels were similar between the vehicle (352 ± 30 pM) and the dapagliflozin group (358 ± 30 pM) at the start of the study. Dapagliflozin significantly reduced blood glucose, although not to the level of non-diabetic mice, and even though blood glucose increased slightly over the study period, the majority of the mice had blood glucose levels below 15 mM for the 8-week study (*p* < 0.0001, 2-way ANOVA, Figure 1C).

### 3.2. SGLT2 Inhibition Preserved Human Insulin Levels in Diabetic Mice

Human C-peptide levels declined over time in the vehicle group, and after 8 weeks, several mice had no detectable human C-peptide, demonstrating a poor or non-functioning islet graft. In the dapagliflozin-treated mice, human C-peptide levels were better preserved (Figure 1D). Similarly, human insulin was low in the control mice but maintained in the dapagliflozin group (Table 1). Mouse C-peptide was low in all alloxan-treated mice, and the pancreatic insulin content was 5%–8% compared to normal mice (Table 1). Interestingly, plasma glucagon was elevated about 5-fold in the alloxan-treated mice (Figure 1E). Following immunohistochemical analysis of the mouse pancreas, we found that the islet area was similar between the three groups (Appendix A). In the normoglycemic sham-operated mice, the pancreatic mouse islets consisted of 96.6% ± 0.7% insulin-positive cells and 3.4% ± 0.7% glucagon-positive cells (Figure 1F, Appendix A), with very strong insulin staining (Figure 1G). However, in the alloxan-diabetic mice, there was a dramatic increase in the glucagon-positive cells representing 55%–60% of the cells while only 40%–45% of the pancreatic islet cells were insulin-positive (Appendix A and Figure 1F,H,I), with no significant difference between vehicle and dapagliflozin-treated mice. The pancreatic insulin staining was extremely weak in the alloxan-treated mice, with no apparent differences between vehicle- (Figure 1H) and dapagliflozin-treated mice (Figure 1I).

Multiple metabolic parameters were measured in plasma and tissue samples at the end of the study with no significant difference between the diabetic controls and the dapagliflozin-treated mice except for blood glucose and plasma fructosamine levels, which were reduced by dapagliflozin treatment (Table 1). However, when compared to the normoglycemic, sham-operated mice, there were significant differences in multiple parameters.

The diabetic mice had increased production of ketone bodies and low levels of liver triglycerides, indicating increased fat oxidation (Table 1). There was also a small increase in plasma cholesterol levels with dapagliflozin, while plasma triglyceride levels were similar in all three groups. There was a small increase in plasma ALT in the diabetic vehicle mice, while haptoglobin was similar between the groups indicating no significant liver inflammation (Table 1).

### 3.3. SGLT2 Inhibition Preserved Human Beta Cell Function

To explore the function of the grafted human islets, the maximal insulin secretory capacity was examined in an intravenous arginine and glucose tolerance test (IVArgGTT). Basal, i.e., two-hour fasted blood glucose, in the alloxan-diabetic mice was significantly lower in the dapagliflozin-treated group compared to the vehicle group (Figure 2A) and was similar to the non-diabetic control group, while the 1 min value after administration of glucose and arginine was not significantly different between the groups. After 10 min, glucose was again lower in the dapagliflozin group compared to the vehicle group. The glucose excursion curve for the normoglycemic mice was overlapping with the dapagliflozin-treated mice (Figure 2A), indicating normal glucose control. The insulin assay detects both human and mouse insulin, and the total insulin levels were higher in the dapagliflozin-treated group at 0 and 1 min compared to the vehicle group but not different to the normoglycemic sham group (Figure 2B). However, calculation of the AUC for both glucose and insulin demonstrated a significant increase in the amount of secreted insulin in relation to the glucose excursions in the dapagliflozin group, indicating preserved beta cell function following dapagliflozin treatment in alloxan-diabetic mice (Figure 2C).

### 3.4. SGLT2 Inhibition Increased Human Alpha and Beta Cell Proliferation and Reduced Apoptosis

Using a triple-staining method for insulin (beta cell marker), glucagon (alpha cell marker), and Ki67 (proliferation marker) and image processing, we determined the percentage of proliferating human alpha and beta cells in relation to the treatment (Figure 3A,B). The average number of insulin and glucagon-positive cells detected in the human islet grafts were similar between vehicle and dapagliflozin-treated mice (a total of 6010 ± 930 vs. 5410 ± 810 insulin or glucagon-positive cells per graft, measured in vehicle vs. dapagliflozin-treated diabetic mice, *p* > 0.05). The percentage of insulin^+^ cells was higher in the dapagliflozin group (Figure 3C), while the glucagon^+^ cells were fewer compared to the controls (Figure 3D). Ki67 staining was used to identify proliferating cells. The number of dual Ki67^+^insulin^+^ cells was significantly higher in the dapagliflozin-treated group compared to the diabetic control mice (Figure 3E). The alpha cells proliferated at a higher rate compared to beta cells, particularly in the dapagliflozin group (Figure 3F).

When exploring the data from the individual mice, we found that the long-term hyperglycemia, calculated as the AUC of the glucose curves during the 8-week study (data shown in Figure 1C), negatively correlated to the percentage of insulin^+^ cells (Figure 4A) and positively correlated with the percentage of glucagon^+^ cells (Figure 4B). Thus, with prolonged hyperglycemia, the insulin^+^ cells were reduced, and instead, the glucagon^+^ cells increased. It is well known that short-term hyperglycemia results in increased beta cell proliferation and increased beta cell mass, while chronic hyperglycemia leads to glucotoxicity and reduced beta cell mass [39,40,41]. In our study, both alpha and beta cell proliferation were low in the severely hyperglycemic mice. However, in the dapagliflozin group, there was no correlation between alpha and beta cell proliferation and glucose control, but several of the mice demonstrated a much higher degree of alpha and beta cell proliferation, detected as Ki67^+^glucagon^+^ or Ki67^+^insulin^+^ cells, respectively (Figure 4C,D). The mice in the dapagliflozin-treated group were divided into two groups based on the islet cell proliferation, one group with higher proliferation (≥1% beta cell proliferation and ≥2.5% alpha cell proliferation) and one group with lower proliferation (<1% beta cell and <2.5% alpha cell proliferation). The proliferation rate in adult human islets is extremely low and has been estimated to be <0.5% [42]. The dapagliflozin-treated mice with high islet cell proliferation were compared to those with lower proliferation. Interestingly, there was no difference in blood glucose between these two groups or any of the other glucocentric parameters (glucose during the IVArgGTT or fructosamine, Appendix A). There was also no significant difference in human C-peptide levels (373 ± 114 vs. 238 ± 84 pM in mice with high proliferating grafts). In the mice with the lower proliferation rate, the percentage of alpha and beta cells was 38% and 60%, respectively, while in the high proliferating group, it was 44% and 49%, suggesting that the altered ratio of alpha to beta cells may play a role for the islet function. Glucagon levels were very high in alloxan-treated mice compared to normal mice but with no difference between the high and low proliferating groups (Appendix A).

Increased apoptosis may contribute to the loss of beta cells in diabetes. To explore the degree of apoptosis, the number of caspase 3-positive cells within the human islet cells was detected. Using a triple-staining method, the number of double insulin^+^caspase3^+^ (purple/brown) or glucagon^+^caspase3^+^ (yellow/brown) cells were counted (Figure 5A,B). There were few caspase 3^+^ cells, indicating a low level of apoptosis in the grafts from both groups. The number of apoptotic insulin^+^ and glucagon^+^ cells was significantly lower in dapagliflozin-treated mice (Figure 5C,D).

During the extreme loss of mouse beta cells, alpha cells can transdifferentiate into beta cells, involving PDX-1 (beta cell marker) co-expression with glucagon (alpha cell marker) [8,15]. To explore whether such cells existed in the transplanted human islets, the grafts were stained for dual expression of PDX-1 and glucagon (Figure 5E,F). Dual PDX-1^+^ and glucagon^+^ cells were found at a low level in many of the islet grafts (observed in the majority of the animals in both groups), suggesting that transdifferentiation (or dedifferentiation) may be taking place in the islet grafts. There was a significant increase in the number of double PDX-1^+^glucagon^+^ cells in the dapagliflozin group compared to the vehicle diabetic control group (Figure 5G).

### 3.5. In Vitro Treatment of Human Islet with Dapagliflozin Had No Effects on Islet Function

After incubation of human islets with dapagliflozin (100 nM) at different glucose concentrations (5.6, 15, and 30 mM) for 96 h incubation, insulin and glucagon secretion were measured. Glucose-stimulated insulin secretion (GSIS) was calculated as the max response at 30 mM over basal (5 mM) glucose. The glucose concentration range used in the in vitro study reflected the concentrations that the human islets were exposed to in the in vivo study. Islets pre-cultured at 5.6 mM glucose had a suitable glucose-stimulated insulin secretion (Appendix A), while culture at high glucose (15 or 30 mM) resulted in significantly elevated basal insulin and a poor GSIS response (Appendix A). GSIS was not changed by dapagliflozin treatment at any glucose concentration. The glucagon response to low glucose was used as a measure of alpha cell function and calculated as the increase in glucagon secretion at 1 mM glucose over basal glucose (5.6 mM). There was no significant difference in the glucagon secretion in response to culture at different glucose concentrations (Appendix A).

## 4. Discussion

The effect of long-term hyperglycemia and treatment with a selective SGLT2 inhibitor, dapagliflozin, for protection of islet function via mechanisms that could go beyond glucose reduction was investigated. Clinical studies show protective effects of SGLT2 inhibitors on human islet function in type 2 diabetes [3,25], and recently, the treatment was shown to be effective and safe also in type 1 diabetes [21,22], while no study has explored changes in islet composition in relation to function and treatment. To that end, we used diabetic mice transplanted with a marginal mass of human islets and treated them with dapagliflozin for 8 weeks. As expected, dapagliflozin treatment resulted in significantly improved glucose homeostasis even in this severely hyperglycemic type 1-like diabetes model. A significant elevation in ketone bodies was observed in the diabetic mice, although with no difference between the dapagliflozin and the vehicle-treated group. The reason for not seeing the classical elevation of ketone bodies with dapagliflozin treatment is most likely due to this type 1 diabetic model, where insulin levels are extremely low, and the use of fat as energy substrate is high. This is reflected by low levels of liver triglycerides in diabetic mice. Analysis of the grafted human islets revealed increased proliferation of both alpha and beta cells in dapagliflozin-treated mice, but despite a high alpha cell proliferation, the number of alpha cells was lower compared to the control mice, implying that the turnover of alpha cells was faster in the dapagliflozin-treated mice. Since it is not possible to trace the origin of the different cells in primary human islets, we performed two more analyses of the grafts to further describe the effect of SGLT2 inhibition on the human islets. First, we explored the degree of apoptosis by detecting caspase 3-positive cells in the graft. The number of apoptotic cells, 8 weeks after transplantation, was very low in both the control and the dapagliflozin-treated group. Still, SGLT2 inhibition significantly reduced apoptosis in both alpha and beta cells. Thus, we found both increased proliferation and decreased apoptosis in both alpha and beta cells after treatment with dapagliflozin.

In the next analysis, we looked for PDX-1 and glucagon double-positive cells as a marker for alpha to beta cell transdifferentiation [8,15,43]. Indeed, double-positive cells were identified in all grafts, and there was a significantly higher number of double-positive cells in the dapagliflozin group compared to the vehicle, suggesting ongoing transdifferentiation in the grafted human islets. Recent studies have shown that insulin and glucagon double-positive cells exist in the human pancreas, suggesting that transdifferentiation is a normal response to poor beta cell glucose responsiveness [43,44].

Beta cell proliferation is known to occur in response to extrinsic factors such as growth factors, prolactin, and glucagon-like peptide 1 (GLP-1) [45]. One study demonstrated increased human beta cell proliferation only in juvenile human islets and not in adult islets following treatment with exendin-4, a GLP-1 receptor agonist, suggesting that the mitogenic signaling is not active in adult islets [42]. We found high human alpha cell proliferation (~1.2% Ki67^+^glucagon^+^) with a significant increase in the dapagliflozin group (~1.8% Ki67^+^glucagon^+^) in adult human islets. Beta cell proliferation rate was augmented with dapagliflozin from ~0.2% to ~0.6%, demonstrating significant effects of dapagliflozin increasing both alpha and beta cell proliferation. One difference between the studies is that we examined alpha and beta cell proliferation in the face of hyperglycemia, which is a factor known to, at least initially, promote beta cell proliferation. However, long-term hyperglycemia will result in declining functional beta cell mass. Our data suggest that in type 1-like diabetes with severe hyperglycemia, SGLT2 inhibition partly protected human islets by increasing the proliferation of both alpha and beta cells and improving the insulin secretory capacity. The significant increase in alpha cell proliferation and the lower number of alpha cells compared to vehicle-treated mice may suggest ongoing transdifferentiation. However, further studies with linage tracing are needed to prove this.

The plasticity of alpha and beta cells has been shown in rodents [13,15,17]. In the healthy mouse pancreas, only a few percent of the islet cells are alpha cells. It is well known that in rodents, alpha cells are located toward the edges of the islets and that the core is filled with beta cells [46]. After alloxan ablation of the mouse beta cells, the area of the pancreatic islets remained the same as in untreated mice, although the number of alpha cells was increased to represent more than 50% of the islet area, and they were no longer located at the edges of the islets, as shown in Figure 1F–I. Thus, ablation of the mouse beta cells resulted in significantly increased alpha cells in the pancreatic islets. Insulin-producing cells were still present, but they contained very little insulin, as shown by the weak insulin staining, the low level of insulin in pancreatic homogenates, and the low levels of circulating mouse C-peptide. Similar changes occurred in the human islets following 8 weeks of hyperglycemia; however, the healthy human islets contained around 30% alpha cells and 70% beta cells, and the organization is different with scattered alpha cells within the islets [46]. In the diabetic state, the human beta cells are lost, and we show that severe hyperglycemia correlates negatively to the beta cell area and positively to the alpha cell area. This demonstrates that hyperglycemia is driving the loss of beta cells while increasing both the alpha cell numbers and function, illustrated by the significantly increased glucagon levels. The increased glucagon secretion may occur due to a lack of optimal regulation when insulin levels are low [46]. Since it is not possible to differentiate between mouse and human glucagon, the contribution of glucagon secretion from the human islets versus the mouse islets cannot be established. However, the importance of the elevated glucagon for alpha to beta cell transdifferentiation was recently illustrated in zebrafish, where transdifferentiation was significantly reduced in fish embryos lacking glucagon [13]. Thus, the elevated glucagon levels in this model may be one driving factor for the observed transdifferentiation of alpha cells into beta cells.

To investigate if dapagliflozin has direct effects in protecting islet function, human islets were incubated for four days with the SGLT2 inhibitor at normal, elevated, and extreme glucose concentrations to mimic the different glucose levels in the in vivo study. The islets kept at normal glucose maintained their normal response to glucose by elevating insulin and reducing glucagon secretion. At higher glucose concentrations, the human islet function deteriorated as expected with high basal insulin secretion and a trending higher glucagon secretion at very low glucose (1 mM). These responses to hyperglycemia are normal, and the addition of dapagliflozin did not change the pattern. The in vitro data do not support a direct effect of dapagliflozin protecting the islet function, which demonstrates that the in vivo situation represents a much more complex pattern of signals responsible for the observed effects of SGLT2 inhibition.

## 5. Conclusions

In summary, SGLT2 inhibition effectively reduced glucose levels in this diabetic, human islet transplantation mouse model. The function of the human islet beta cells was improved in terms of maintaining insulin secretion, and SGLT2 inhibition promoted the proliferation of both alpha and beta cells. While further studies are needed to fully outline the role of SGLT2 inhibition on the preservation of human islet function, this is the first study to explore the effect of SGLT2 inhibition on human islets in vivo. We demonstrate maintained beta cell function with a potential effect of SGLT2 inhibition promoting alpha to beta cell transdifferentiation.

## Figures and Tables

**Figure 1 biomedicines-10-00203-f001:**
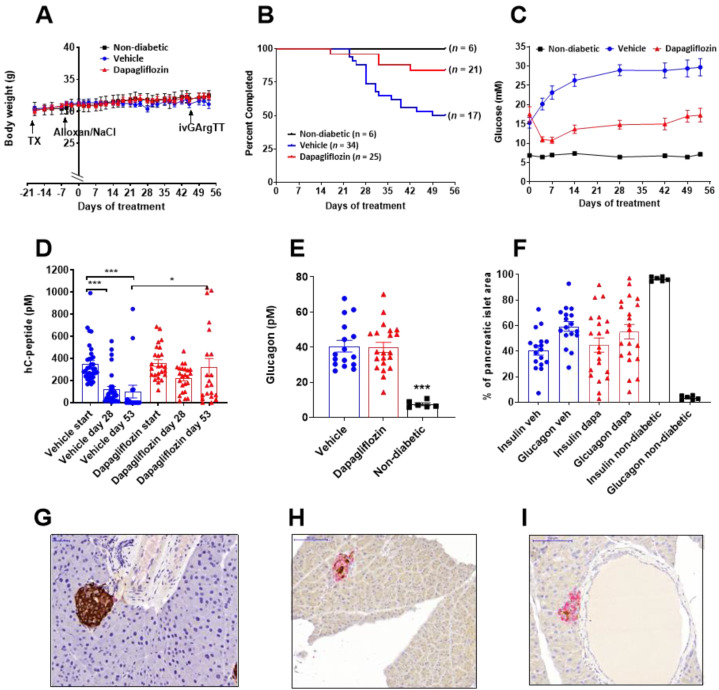
Dapagliflozin increased survival and protected human beta cells in diabetic mice (**A**) Body weight was measured regularly throughout the study. The mice were transplanted (TX) with ~1500 human islet equivalents two weeks before alloxan ablation of the mouse beta cells. Dapagliflozin was given in the drinking water for 8 weeks starting day 0. The diabetic control mice were given tap water. Sham-operated mice with no transplant and no alloxan treatment were used as normoglycemic controls. (**B**) Survival plot over the 8-week study. A total of 59 mice were transplanted with human islets, and 38 mice completed the study. (**C**) Fed glucose values and (**D**) human C-peptide levels measured before dapagliflozin treatment start and after 28 and 53 days. (**E**) Plasma glucagon levels measured at termination. (**F**) Distribution of alpha cells and beta cells in the mouse pancreas in the different treatment groups. Mouse pancreatic islet stained for insulin (brown) and glucagon (purple) in (**G**) normoglycemic control mice, (**H**) vehicle-treated alloxan-diabetic mice, and in (**I**) dapagliflozin-treated alloxan-diabetic mice. The data are presented as mean, and error bars represent SEM. * *p* < 0.05, *** *p* < 0.001. Statistical difference was determined using one-way ANOVA followed by Tukey’s post-hoc test and two-way ANOVA for repeated measures. Number of samples were *n* = 21 (dapagliflozin, red symbols), *n* = 17 (vehicle, blue symbols) and *n* = 6 (sham, black symbols) representing animals that completed the study.

**Figure 2 biomedicines-10-00203-f002:**
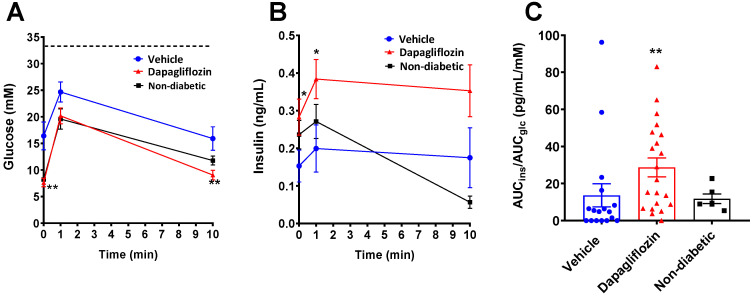
Dapagliflozin improved the human islet response to glucose and arginine. Intravenous arginine and glucose tolerance tests were performed on day 53 in the study. Mice were challenged with 2 g/kg glucose and 0.25 g/kg arginine. (**A**) Glucose excursions during the first 10 min after intravenous administration of glucose and arginine with (**B**) corresponding insulin (human and mouse) levels. (**C**) Area under the curves (AUC) for glucose and insulin were calculated, and the ratio of AUC_ins_ over AUC_glc_ is presented. The dotted line in A represents the highest glucose value that can be measured with the glucometer. Glucose and insulin excursions were analyzed with two-way ANOVA followed by Bonferroni post-hoc test. * *p* < 0.05, ** *p* < 0.01. Number of samples were *n* = 21 (dapagliflozin, red symbols), *n* = 17 (vehicle, blue symbols) and *n* = 6 (sham, black symbols).

**Figure 3 biomedicines-10-00203-f003:**
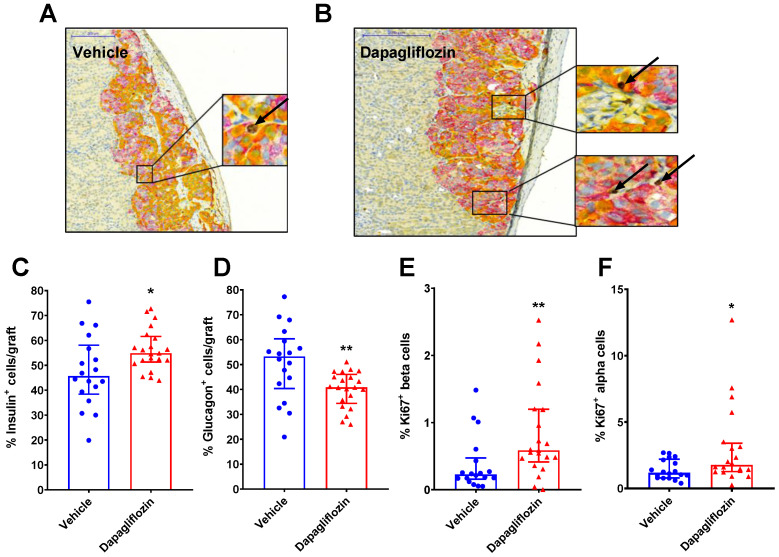
Dapagliflozin increased both alpha and beta cell proliferation. Triple staining of human islets grafted under the kidney capsule showing an example from (**A**) vehicle and (**B**) dapagliflozin-treated mice. Insulin-positive cells are stained in purple, glucagon in yellow, and Ki67 in brown. The arrows indicate the brown Ki67-positive cells. (**C**) Quantification of the insulin-positive cells and (**D**) the positive glucagon cells in human islet grafts. The area of the graft was determined by a computerized model, and the number of insulin and glucagon-positive cells were counted. The analysis also determined the percentage of proliferating beta cells (**E**) insulin^+^ and Ki67^+^ dual-positive cells and alpha cells (**F**) glucagon^+^ and Ki67^+^ dual-positive cells in the grafts. The data are presented as median with error bars representing the interquartile range. Statistical differences were determined with Mann–Whitney’s non-parametric test. * *p* < 0.05, ** *p* < 0.01. Number of samples were *n* = 21 (dapagliflozin, red symbols) and *n* = 17 (vehicle, blue symbols).

**Figure 4 biomedicines-10-00203-f004:**
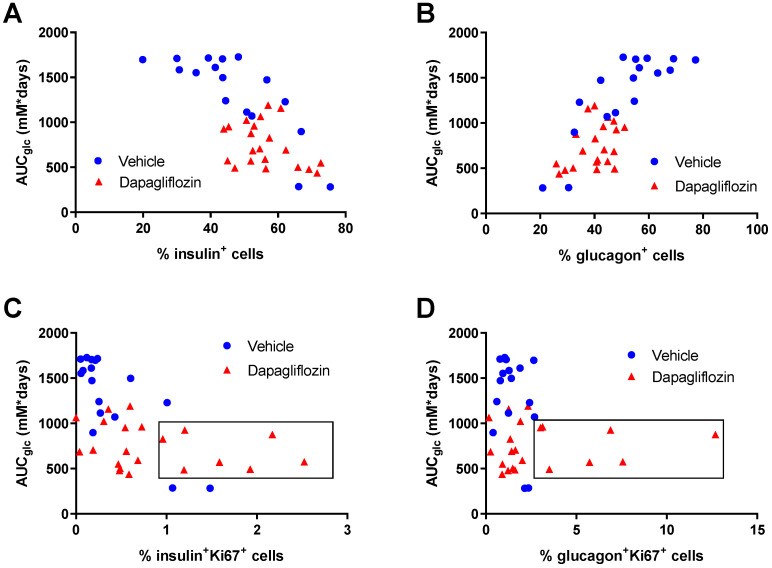
Human islet composition and proliferation in relation to glycemia. Individual analysis of plasma glucose over the 8-week study and the number of insulin and glucagon-positive cells at the end of the study. The AUC for glucose during the study was calculated from the data in Figure 1C and plotted against the percentage of (**A**) insulin-positive and (**B**) glucagon-positive cells or percentage of proliferating (**C**) beta cells (insulin^+^ Ki67^+^) and (**D**) alpha cells (glucagon^+^Ki67^+^) in the human islet grafts. The boxes in (**C**) and (**D**) indicate the group of mice with high alpha and beta cell proliferation rates (>2.5% and >1%, respectively). Number of samples were *n* = 21 (dapagliflozin, red symbols) and *n* = 17 (vehicle, blue symbols).

**Figure 5 biomedicines-10-00203-f005:**
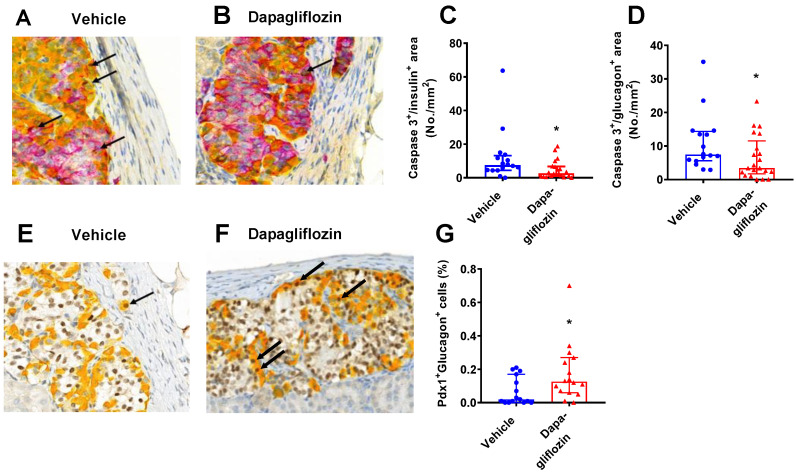
Dapagliflozin reduced alpha and beta cell apoptosis. Triple staining of the human islets grafts with insulin (purple), glucagon (yellow), and caspase 3 (brown) in (**A**) vehicle or (**B**) dapagliflozin-treated mice. The arrows indicate dual Ki67^+^ and insulin^+^ or glucagon^+^ cells. A computerized method was used to count the number of apoptotic (**C**) insulin^+^ cells and (**D**) glucagon^+^ cells. Dual staining for glucagon (yellow) and PDX-1 (brown) in (**E**) vehicle and (**F**) dapagliflozin-treated mice. (**G**) The percentage of dual PDX-1 and glucagon cells was counted. The data are presented as median with error bars representing the interquartile range. Statistical differences were determined with Mann–Whitney’s non-parametric test. * *p* < 0.05. Number of samples were *n* = 21 (dapagliflozin, red symbols), *n* = 16 (vehicle, blue symbols) for caspase 3 analysis and *n* = 16 (dapagliflozin, red symbols), *n* = 15 (vehicle, blue symbols) for PDX1 analysis.

**Table 1 biomedicines-10-00203-t001:** Plasma biochemistry, liver triglyceride, and pancreas insulin content in mice from normoglycemic sham-operated mice, diabetic vehicle mice, and dapagliflozin-treated diabetic mice at the termination of the study.

	Non-Diabetic (*n* = 6)	Diabetic Vehicle (*n* = 21)	Diabetic Dapagliflozin (*n* = 17)
Fructosamine (µM)	220 ± 2	278 ± 13 *	236 ± 8 #
β-hydroxybutyrate (µM)	143 ± 15	306 ± 29 *	300 ± 38 *
Liver triglycerides (g/100 g tissue)	0.97 ± 0.15	0.33 ± 0.029 ***	0.44 ± 0.068 ***
Cholesterol (mM)	2.09 ± 0.04	2.28 ± 0.064	2.45 ± 0.058 **
Triglycerides (mM)	1.39 ± 0.18	1.69 ± 0.23	1.54 ± 0.16
ALT (µkat/L)	0.44 ± 0.03	0.60 ± 0.031 *	0.54 ± 0.034
Haptoglobin (g/L)	1.32 ± 0.29	0.44 ± 0.07	1.01 ± 0.30
Human insulin (ng/mL)	ND	0.055 ± 0.034	0.25 ± 0.072 #
Mouse C-peptide (ng/mL)	0.67 ± 0.1	0.05 ± 0.01 ***	0.06 ± 0.01 ***
Glucagon (pM)	7.5 ± 0.8	40.6 ± 3.2 ***	39.8 ± 2.9 ***
Pancreatic insulin (ng/µg protein)	6.3 ± 0.4	0.11 ± 0.02 ***	0.24 ± 0.0.07 ***

The results are presented as mean ± SEM, and statistical significance was determined using one-way ANOVA, followed by Tukey’s post-hoc test for differences between the groups. * *p* < 0.05, ** *p* < 0.01, *** *p* < 0.001 for significant difference compared to the normoglycemic sham group, # *p* < 0.05 vehicle compared to the dapagliflozin-treated mice. ALT: alanine transaminase, TG: triglyceride, ND: not determined.

## Data Availability

The data supporting this study can be made available upon sending a request to the corresponding author.

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
