# Peer review of "Inhibition of SGLT2 Preserves Function and Promotes Proliferation of Human Islets Cells In Vivo in Diabetic Mice"

_biomedicines, 2022, doi:10.3390/biomedicines10020203_

Round 1

Reviewer 1 Report

The study by Karlsson and coll. examines effects of dapagliflozin on human islets in vivo and in vitro. In vitro treatment of human islets with dapagliflozin had no impact on islet function. In vivo, SGLT2 inhibition supported human islet function and promoted alpha to beta cell transdifferentiation. If renal and cardiovascular impact are well described, a possible evidence for human islet cell plasticity with this glucose lowering drug is of course interesting not only from a physiological point of view but also from a clinical point of view (delay insulin in Type 2).Furthermore, this study undelines the importance of conducting pharmacoogical studies with primary human islets to better predict the clinical outcome. I have the following questions to the authors : - Fig1 A: Could you comment on stable body weight with dapa/vehicle/controls ? - Increased production of ketone bodiesin dapa treated mice (Table 1). This confirms the classical clinical side effect. This finding should be discussed.

Author Response

Dear reviewer,

Thank you for your comments on our manuscript. We have the following answers: 

- Fig1 A: Could you comment on stable body weight with dapa/vehicle/controls ?

These mice were 8 weeks old and adult at the start of the study. At 8 weeks of age the mouse model is described to have a body weight of about 30 g. Over the 8 week study body weight increased slightly in all three groups with no difference between treatment groups with drug or even with the control that received no drug and no transplant. This shows that the mice in all groups that completed the study were in good condition maintaining their body weight.

Since in severe type 1 diabetes, body weight loss is a clinical sign, mice that could not maintain their body weight were terminated and removed from the study (according to the criteria for the study and the ethical permission). See section 2.1.  

- Increased production of ketone bodies in dapa treated mice (Table 1). This confirms the classical clinical side effect. This finding should be discussed.

In type 1 diabetes, ketone bodies increase due to lack of sufficient glycogen stores and high turnover of fat in the adipose tissue. It is also described that treatment with an SGLT2 inhibitor increase the ketones as the reviewer elutes to. There was however, no difference in ketone body levels between the vehicle and the dapagliflozin treated group (Table 1), most likely due to the diabetic status and elevated blood glucose levels in the animals. Thus, the higher ketone bodies can be considered as a demonstration of the metabolic shift that occurs when either sugar is excreted in the urine and the body switch to use fatty acids as their source of energy or as a result of low insulin and poor glucose metabolism in the type 1 diabetic state. This can be added to the discussion.

Reviewer 2 Report

The article is well-written and informative. I have the following comments...

1. Even if Dapagliflozon is a well know sodium-glucose cotransporter inhibitor, it is important to show if the drug indeed caused the inhibition. 

I recommend the authors show SGLT2 inhibition upon dapagliflozon in both in vivo and in-vitro experiments. 

2. 30mM glucose is not physiological. I recommend 5mM and 15mM data be included  in the manuscript and the 30mM data be removed?

3. In addition to insulin(ß cell marker) and glucagon(α cell marker) and ki67, are there redundant measures/markers that can be shown as evidence for transdifferention?

4. What percentage of the transplant was integrated into the host tissue? and how was that confirmed?

Author Response

Dear reviewer,

Many thanks for your comments on our manuscript. We have the following response to the questions:

  1. Even if Dapagliflozon is a well know sodium-glucose cotransporter inhibitor, it is important to show if the drug indeed caused the inhibition. 

I recommend the authors show SGLT2 inhibition upon dapagliflozon in both in vivo and in-vitro experiments. 

Since dapagliflozin is a registered drug there is always a quality release of the drug substance before it is being used in any experiments. That the drug is active is also clearly shown in the in vivo experiments since the blood glucose was significantly reduced after dosing and there was an increased urine volume. This was not measured but noted in the bedding material being more wet. We have in other experiments studied the effect of in dapagliflozin in increased urine production and elevation of urine glucose. This was however not done in the present study. The drug that was used in the in vitro experiments is the same as was used in the in vivo experiments. 

We can provide information about the activity of the drug when it was released but there is no possibility to test the solutions that was used in the study. Having the typical in vivo effects observed we are confident that we used active substance in the study. We also measured the concentration of the drug and detected clinical relevant concentration in the circulation with the dose that was administered. 

  1. 30mM glucose is not physiological. I recommend 5mM and 15mM data be included  in the manuscript and the 30mM data be removed?

We agree with the reviewer that the 30 mM glucose is not a physiological glucose level and the reason for having it included in the in vitro studies was to compare with the severely diabetic mice that indeed had 30 mM in blood sugar. We propose to keep the results for 30 mM glucose in the in vitro study but add the explanation as to why it was included.

  1. In addition to insulin(ß cell marker) and glucagon(α cell marker) and ki67, are there redundant measures/markers that can be shown as evidence for transdifferention?

We present data for dual glucagon/Pdx1 staining as a measure of ongoing transdifferentiation. Glucagon is a marker for alpha cells while pdx1 is a marker for beta cells. Having both these genes present in the same cells indicate transition between the two cell types. We see the co-localization occurring also in non-treated animals which suggest this process occurs normally, although to a higher extent with SGLT2 inhibition. As stated in the manuscript we need more experiments to establish the mechanism behind the transdifferentiation of islet cells.

We have in this study used as many markers as we could to study the grafts with IHC and due to the limited size of the grafted islets it was not possible to include more markers in this study.  

  1. What percentage of the transplant was integrated into the host tissue? and how was that confirmed?

This is very difficult to determine. The surgical procedure is done by injecting a small volume of PBS (10 µl) containing the islets under the kidney capsule. The islet graft is visible and we use the circulating human insulin levels to judge if the graft is active and integrated. The human insulin levels did not significant differed between the vehicle and the dapa group prior to start of the treatment.

Round 2

Reviewer 2 Report

The glucose concentration range used 
in the in vitro study were reflected the concentrations that the human islets were exposed to in the in vivo study. 

Please remove "were"

Author Response

The sentence has been corrected.